# First Molecular Detection and Genotype Identification of *Toxoplasma gondii* in Chickens from Farmers’ Markets in Fujian Province, Southeastern China

**DOI:** 10.3390/pathogens12101243

**Published:** 2023-10-13

**Authors:** Meng-Jie Chu, Li-Yuan Huang, Wen-Yuan Miao, Ya-Fei Song, Ying-Sheng Lin, Si-Ang Li, Dong-Hui Zhou

**Affiliations:** 1Key Laboratory of Fujian-Taiwan Animal Pathogen Biology, College of Animal Sciences (College of Bee Science), Fujian Agriculture and Forestry University, Fuzhou 350002, China; mengjiechu@163.com (M.-J.C.); huangliyuan0129@163.com (L.-Y.H.); wenyuan19940704@163.com (W.-Y.M.); s17759786286@163.com (Y.-F.S.); 2Zhangzhou Animal Husbandry Technical Service Station, Zhangzhou 363000, China; linyingsheng2-10@163.com

**Keywords:** *Toxoplasma gondii*, Fujian province, chicken, prevalence, genotype

## Abstract

*Toxoplasma gondii* is an opportunistic pathogenic protozoan that can infect all nucleated cells in almost all warm-blooded animals, including humans. *T. gondii* infection has been reported in many food animals worldwide. However, the prevalence and genotypes of *T. gondii* in chickens from farmers’ markets in Fujian province in southeastern China remain unreported. In the present study, four tissue samples from each of the 577 chickens (namely, the heart, liver, lungs, and muscles) were collected from farmers’ markets in five regions of Fujian province (Zhangzhou, Sanming, Quanzhou, Fuzhou, and Longyan). We first analyzed the prevalence and genotypes of *T. gondii* using PCR targeting of the *B1* gene of *T. gondii*. Of the 577 chickens, thirty-two (5.5%) tested positive for the *B1* gene. Among the five regions, Sanming had the highest infection rate (16.8%, 16/95), followed by Quanzhou (8.0%, 8/100), Longyan (5.0%, 5/100), Zhangzhou (1.1%, 2/182), and Fuzhou (1.0%, 1/100). Among these thirty-two *T. gondii*-positive chickens, the infection rates of the lungs, heart, liver, and muscles were 68.8% (22/32), 34.4% (11/32), 28.1% (9/32), and 9.4% (3/32), respectively. Significant differences in prevalence were found among the different regions (*χ*^2^ = 35.164, *p* < 0.05) and tissues (*χ*^2^ = 25.874, *p* < 0.05). A total of 128 tissue and organ samples of the thirty-two *T. gondii*-positive chickens from the different regions were analyzed using PCR–restriction fragment length polymorphism (PCR–RFLP) on the basis of 10 genetic markers. Seven tissue samples (lung samples from five chickens, heart samples from one chicken, and liver samples from one chicken) underwent successful amplification at all the genetic markers, and all the *T. gondii* genotypes were identified as genotype I (ToxoDB #10). These findings serve as a foundation for evaluating the risk of *T. gondii* contamination in chicken products intended for human consumption and offer insight into preventing the transmission of the parasite from chickens to humans.

## 1. Introduction

*Toxoplasma gondii* is an opportunistic pathogen that can parasitize in the nucleated cells of virtually every warm-blooded animal, including humans and chickens [1,2]. Healthy individuals infected with *T. gondii* may be asymptomatic. Infection in pregnant women can lead to premature birth, miscarriage, stillbirth, and fetal developmental malformations [3]. Acute toxoplasmosis retinochoroiditis can cause pain, photophobia, tearing, and loss of vision [4]. In addition, *T. gondii* infection can occur through organ transplantation, accidental inoculation of tachyzoites in the skin, blood transfusions, and the transplacental route [5,6,7]. The major microscopic lesions observed in chickens infected with *T. gondii* include sciatica, chorioretinitis, encephalitis, myocarditis, and pericarditis, highlighting the significant pathological impact of infection [8].

Foodborne illness and food contamination have emerged as serious and progressively worsening global public health problems. However, different food sources, including unpasteurized milk, fresh produce, and undercooked or raw meat, serve as vehicles for the transmission of toxoplasmosis to humans [6]. Felines play an important role as the final host of *T. gondii* infection [2,9]. The shedding of oocysts in cat feces can lead to environmental contamination [5]. Humans are infected with *T. gondii* by ingesting food and water contaminated with *T. gondii* oocysts or by consuming raw or undercooked meat containing *T. gondii* cysts [10,11,12,13]. Currently, the commonly used genotyping methods for *T. gondii* include PCR–restriction fragment length polymorphism (PCR–RFLP), the microsatellite typing method, and multilocus sequence typing (MLST). *B1* genes are frequently used as targets for screening *T. gondii* infections due to their high level of sensitivity [14].

Chickens, as an intermediate host of *T. gondii*, usually exhibit chronic infections without obvious clinical symptoms [7,15]. Free-range chickens are important in the epidemiology of *T. gondii* because they often forage from the ground and are susceptible to soil contamination in the environment. Their infection with *T. gondii* is considered an indicator of the level of environmental contamination [15]. The overall seroprevalence of *T. gondii* IgG antibody was 9.38% among 1045 neonates in Fujian Province, China, according to Wu et al. in 2020, but the prevalence and genotypes of *T. gondii* in chickens in Fujian Province have not been reported [16]. Therefore, this study conducted an epidemiological survey of *T. gondii* infection in chickens from farmers’ markets in five regions of Fujian province using nested PCR and identified *T. gondii* genotypes using PCR–RFLP. This study provides an initial assessment of the transmission risk of toxoplasmosis in edible chickens and offers crucial information for preventing the transmission of this parasite from animals to humans, thereby safeguarding public health.

## 2. Materials and Methods

### 2.1. Geographical Features of Fujian

Fujian province, situated on the southeast coast of China, is geographically located within 23°31′–28°18′ N latitude and 115°50′–120°43′ E longitude (Figure 1). It faces Taiwan across the water and holds significant ecological importance as a crucial stopover destination for migratory birds from Australia. Furthermore, it serves as a pivotal hub for the staging, wintering, and breeding endeavors of diverse migratory bird species, underscoring its indispensable role in avian ecology.

### 2.2. Sample Collection

A total of 577 free-range chickens were purchased from farmers’ markets in five regions (182 from Zhangzhou, 95 from Sanming, 100 from Quanzhou, 100 from Fuzhou, and 100 from Longyan) of Fujian province. The heart and parts of the liver, lungs, and muscles were collected from each chicken. Approximately 100 g of each sample was meticulously collected and individually placed in sterile polythene bags, which were then tightly sealed to maintain their sterility and integrity. The sampling time, location, and other relevant sample information were recorded. The collected tissue samples were stored at –20 °C for further testing.

### 2.3. Extraction and Detection of Tissue DNA

A piece of each sample (including 577 heart, 577 liver, 577 lung, and 577 muscle samples) weighing approximately 100 mg was randomly cut for DNA extraction using a commercial E.Z.N.A.^®^ Tissue DNA kit (Omega Biotek Inc., Norcross, GA, USA) according to the manufacturer’s protocol. PCR was performed to detect infection with *T. gondii*, targeting the *B1* gene. The final amplified products were subsequently subjected to electrophoresis in agarose gel stained with Gold ViewTM, and their visualization was achieved under UV light.

### 2.4. Genetic Characterization of T. gondii

PCR–RFLP analysis of *T. gondii* samples was conducted using 10 different genetic markers (SAG1, 5′–SAG2, 3′–SAG2, alt. SAG2, SAG3, BTUB, GRA6, c22-8, c29-2, L358, PK1, and Apico). *T. gondii* reference strains (GT1, PTG, CTG, MAS, TgCgCa1, TgCatBr5, TgCatBr64, and TgToucan (TgRsCr1)) were used as positive controls. Nested PCR products from each marker were digested with selected restriction enzymes, and DNA fragments were isolated from agarose. All markers were separated in 2.5% gels, except Apico, which was resolved in 3% gel. Lastly, the genotypes of the *T. gondii* isolates were distinguished, and the results were determined on the basis of the *T. gondii* database (www.toxodb.org, accessed on 7 July 2023). The primers and restriction enzymes used in this study were described previously [17].

### 2.5. Data and Statistical Analysis

The prevalence of *T. gondii* infection was determined according to the following formula:

Prevalence of infected chickens (%) = 100 × number of PCR-positive chickens regarding at least one tissue/total number of tested chickens.

SPSS25.0 software was used for statistical analysis. The chi-square test (χ^2^) was used to compare the *T. gondii* infection rates in chickens from different regions and tissues in Fujian province, with *p <* 0.05 indicating significant difference.

## 3. Results

In this study, heart, liver, lung, and muscle tissue samples from 577 farmers’ market chickens from five areas in Fujian province were examined using PCR based on the *B1* gene. The results showed that the infection rate of *T. gondii* in chickens was 5.5% (32/577). In different areas, Sanming had the highest infection rate (16.8%, 16/95), followed by Quanzhou (8.0%, 8/100), Longyan (5.0%, 5/100), Zhangzhou (1.1%, 2/182), and Fuzhou (1.0%, 1/100). Statistical analysis showed significant differences in the infection rate of *T. gondii* among the different regions (χ^2^ = 35.164, *p* < 0.05) (Table 1).

We performed statistical analyses of multiple tissues and organs from *T. gondii*-positive chickens according to different regions. The PCR results of 128 tissue and organ samples of the thirty-two *T. gondii*-positive chickens from different regions showed the presence of multiple tissues or organs infected in the same animal, with one chicken being positive in all four tissues. In the different tissues and organs of the thirty-two positive chickens, the lungs (68.8%, 22/32) had the highest infection rate, followed by the heart (34.4%, 11/32), liver (28.1%, 9/32), and muscles (9.4%, 3/32). There was a significant difference in the prevalence of *T. gondii* infection among the different tissues (χ^2^ = 25.874, *p* < 0.05) (Table 2).

A total of 128 tissue and organ samples were collected from the thirty-two *T. gondii*-positive chickens and genotyped using 10 genetic markers. The results found that lung samples of five chickens from Quanzhou (QZC68), Zhangzhou (ZZC18) and Sanming (SMC20, SMC22 and SMC30), one heart sample of chicken from Zhangzhou (ZZC19) and one liver sample of chicken from Sanming (SMC31) were successfully amplified at 10 genetic markers, respectively (Figure 2). The genotypes were all identified as genotype I (ToxoDB #10) (Table 3).

## 4. Discussion

Chickens are becoming increasingly prominent in the dietary structure of Chinese residents and are a major meat product. Toxoplasmosis is widely prevalent across the world. Enhancing the detection of *T. gondii* in chicken products is essential to ensure food safety and public health [18]. PCR is a specific, rapid, sensitive, and cost-effective method for detecting *T. gondii* in chickens [17].

In this study, the total infection rate of *T. gondii* in chickens was 5.5% (32/577), which is higher than the infection rate of *T. gondii* in caged chickens in Gansu province (3.2%) [19], but lower than that in chickens in other regions, such as Shandong province (12.34%) [20], India (6.06%) [21], Iran (8%) [22], Brazil (42%) [23], Egypt (47.2%) [24], and Kenya (79%) [25]. The infection rates of *T. gondii* in chickens from high to low in the five sampled regions were Sanming (16.8%, 16/95), Quanzhou (8.0%, 8/100), Longyan (5.0%, 5/100), Zhangzhou (1.1%, 2/182), and Fuzhou (1.0%, 1/100). Significant differences in prevalence were found among the different regions. This may have been because of the choice of detection method employed during the epidemiological investigation; the geographical location itself; seasonal factors such as precipitation, temperature, and air pressure; specific farm feeding modes; and other factors [26]. Notably, free-range chickens are at higher risk of *T. gondii* infection due to their proximity to soil, plants, or water sources contaminated by *T. gondii* oocysts. This high susceptibility is attributed to their independent foraging behavior, which includes feeding on insects, earthworms, and plants. Moreover, free-range households often keep cats for rodent prevention, which may increase the risk of *T. gondii* infection in free-range chickens [27,28].

*T. gondii* infection is not isolated to single cases in chicken samples. Khan et al. reported that the prevalence of *T. gondii* in chickens was higher in the liver (10.5%) than in the heart (9.5%) and muscle (7.11%) in Pakistan [29]. Zrelli et al. detected the presence of DNA of *T. gondii* in the breast muscle, thigh muscle, heart, and gizzard of free-range chickens in Tunisia through PCR, with the results showing that the hearts (48.3%, 29/60) had the highest infection rate [30]. The aforementioned research highlights that the livers and hearts of chickens are particularly susceptible to *T. gondii* infection. These organs can be considered as preferred sites for diagnostic sampling and evaluation of *T. gondii* infection in chickens. The present study found that the lungs had a higher infection rate. Most of the internal organs of chickens (such as the heart and liver) are consumed by humans, but the lungs are not as popular for consumption as food. Contaminated lungs may be casually discarded by humans. This increases the risk of the transmission of *T. gondii* to humans, stray cats, pigs, other avians, etc. The experimental samples collected in this study were from free-range chickens at the farmers’ markets. They feed mainly from the ground and are susceptible to *T. gondii* [15], which could have led to an increased risk of *T. gondii* infection in individuals who consumed the chicken meat. In addition, slaughterhouse workers and meat sellers who come into contact with raw meat and animal offal may be at greater risk than the general population [31].

In this study, seven tissue samples completed amplification of all the genetic markers, and all the *T. gondii* genotypes were identified as genotype I (ToxoDB #10). Genotype I has been detected in a variety of animal hosts in China, including sheep, Plateau pikas, humans, cats, pigs, bats, black goats, microtus fortis, tree sparrows [32,33,34,35,36,37,38,39,40,41], etc., indicating that genotype I is widely distributed among species in different regions in China. Similar results have been obtained for free-range chickens in Brazil and Colombia, and most of these isolates were genotype I [42,43]. Type I was reported to predominate in retail meats in the UK, Brazil, and Iran [44,45,46]. These findings suggest that the *T. gondii* genotype I can be transmitted between different hosts in different regions. Whether this genotype is widespread in other animals in Fujian province, further studies are necessary for *T. gondii* genotypes in other hosts in this area.

## 5. Conclusions

In conclusion, the total infection rate for *T. gondii* was 5.5% (32/577) in chickens in Fujian province, southeastern China, and all the *T. gondii* genotypes were identified as genotype I (ToxoDB #10). This study provides a basic assessment of the risk of *T. gondii* infection via edible chicken and provides a certain reference value for the development of the poultry industry and guaranteeing human public health security in Fujian province.

## Figures and Tables

**Figure 1 pathogens-12-01243-f001:**
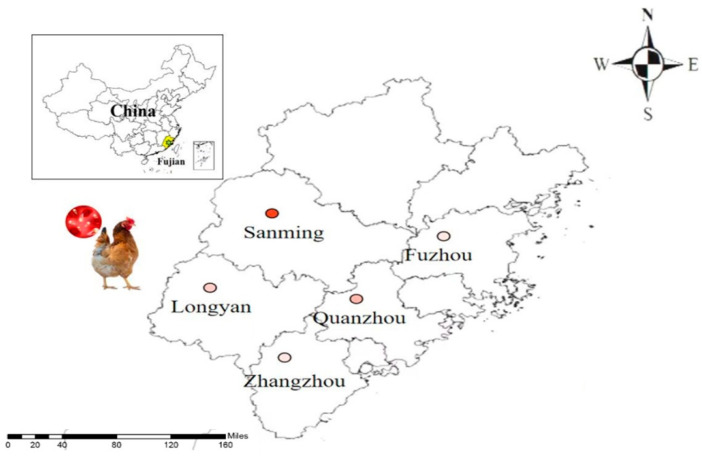
Geographic distribution of the sampling sites in Fujian province, southeastern China.

**Figure 2 pathogens-12-01243-f002:**
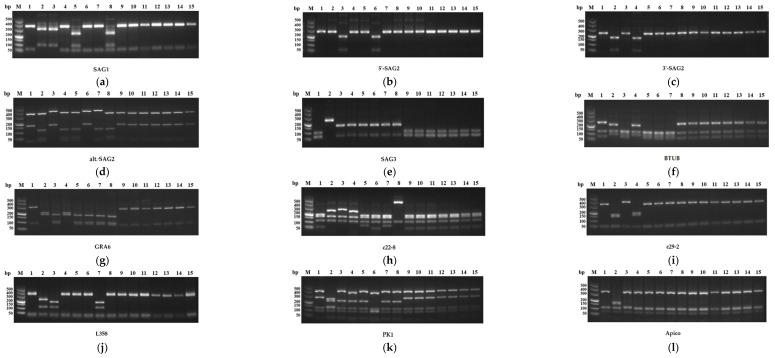
PCR–RFLP analysis of *T. gondii* isolates using 10 different genetic markers. (**a**–**l**) PCR–RFLP polymorphism cleavage map of the 10 different genetic markers (SAG1, 5′–SAG2, 3′–SAG2, alt. SAG2, SAG3, BTUB, GRA6, c22-8, c29-2, L358, PK1, and Apico) of the chicken *T. gondii* isolates in Fujian province. Numbers 1–8 denote the standard strains GT1, PTG, CTG, TgCgCa1, MAS, TgCatBr5, TgCatBr64, and TgToucan (TgRsCr1), respectively; numbers 9–15 denote the samples ZZC18 (lung), ZZC19 (heart), SMC20 (lung), SMC22 (lung), SMC30 (lung), SMC31 (liver), and QZC68 (lung), respectively.

**Table 1 pathogens-12-01243-t001:** The prevalence of *T. gondii* infection in chickens from different areas of Fujian province.

Areas	No. Tested	No. Positive	Infection Rate
Zhangzhou	182	2	1.1%
Sanming	95	16	16.8%
Quanzhou	100	8	8.0%
Fuzhou	100	1	1.0%
Longyan	100	5	5.0%
Total	577	32	5.5%

**Table 2 pathogens-12-01243-t002:** Prevalence of *T. gondii*-positivity in chickens in the different tissues and organs.

Areas	Postive/Tissue	Tissue Type			
		Heart	Liver	Lung	Muscle
Zhangzhou	2 (2 × 4)	2/2	0/2	1/2	0/2
Sanming	16 (16 × 4)	4/16	6/16	11/16	0/16
Quanzhou	8 (8 × 4)	2/8	3/8	5/8	3/8
Fuzhou	1 (1 × 4)	0/1	0/1	1/1	0/1
Longyan	5 (5 × 4)	3/5	0/5	4/5	0/5
Total	32 (32 × 4)	11/32 (34.4%)	9/32 (28.1%)	22/32 (68.8%)	3/32 (9.4%)

Note: Multiple tissue infections exist in the same chicken.

**Table 3 pathogens-12-01243-t003:** PCR–RFLP genotyping of the *T. gondii* isolates of chickens from Fujian province.

Strain Designation	Host	Area	SAG1	5′-SAG2	3′-SAG2	alt. SAG2	SAG3	BTUB	GRA6	c22-8	c29-2	L358	PK1	Apico	Genotypes
GT1	Goat	United States	Ⅰ	Ⅰ	Ⅰ	Ⅰ	Ⅰ	Ⅰ	Ⅰ	Ⅰ	Ⅰ	Ⅰ	Ⅰ	Ⅰ	Reference, Type Ⅰ, ToxoDB #10
PGT	Sheep	United States	II/III	II	II	II	II	II	II	II	II	II	II	II	Reference, Type II, ToxoDB #1
CTG	Cat	United States	II/III	III	III	III	III	III	III	III	III	III	III	III	Reference, Type III, ToxoDB #2
TgCgCa1	Cougar	Canada	Ⅰ	II	II	II	III	II	II	II	u-1	Ⅰ	u-2	Ⅰ	Reference, ToxoDB #66
MAS	Human	France	u-1	Ⅰ	Ⅰ	II	III	III	III	u-1	Ⅰ	Ⅰ	III	Ⅰ	Reference, ToxoDB #17
TgCatBr5	Cat	United States	Ⅰ	III	III	III	III	III	III	Ⅰ	Ⅰ	Ⅰ	u-1	Ⅰ	Reference, ToxoDB #19
TgCatBr64	Cat	Brazil	Ⅰ	Ⅰ	Ⅰ	u-1	III	III	III	u-1	Ⅰ	III	III	Ⅰ	Reference, ToxoDB #111
TgRsCr1	Toucan	Costa Rica	u-1	Ⅰ	Ⅰ	II	III	Ⅰ	III	u-2	Ⅰ	Ⅰ	III	Ⅰ	Reference, ToxoDB #52
ZZC18 (lung)	Chicken	Fujian, China	I	Ⅰ	Ⅰ	Ⅰ	Ⅰ	Ⅰ	Ⅰ	Ⅰ	Ⅰ	Ⅰ	Ⅰ	Ⅰ	ToxoDB #10
ZZC19 (heart)	Chicken	Fujian, China	Ⅰ	Ⅰ	Ⅰ	Ⅰ	Ⅰ	Ⅰ	Ⅰ	Ⅰ	Ⅰ	Ⅰ	Ⅰ	Ⅰ	ToxoDB #10
SMC20 (lung)	Chicken	Fujian, China	Ⅰ	Ⅰ	Ⅰ	Ⅰ	Ⅰ	Ⅰ	Ⅰ	Ⅰ	Ⅰ	Ⅰ	Ⅰ	Ⅰ	ToxoDB #10
SMC22 (lung)	Chicken	Fujian, China	Ⅰ	Ⅰ	Ⅰ	Ⅰ	Ⅰ	Ⅰ	Ⅰ	Ⅰ	Ⅰ	Ⅰ	Ⅰ	Ⅰ	ToxoDB #10
SMC30 (lung)	Chicken	Fujian, China	Ⅰ	Ⅰ	Ⅰ	Ⅰ	Ⅰ	Ⅰ	Ⅰ	Ⅰ	Ⅰ	Ⅰ	Ⅰ	Ⅰ	ToxoDB #10
SMC31 (liver)	Chicken	Fujian, China	Ⅰ	Ⅰ	Ⅰ	Ⅰ	Ⅰ	Ⅰ	Ⅰ	Ⅰ	Ⅰ	Ⅰ	Ⅰ	Ⅰ	ToxoDB #10
QZC68 (lung)	Chicken	Fujian, China	Ⅰ	Ⅰ	Ⅰ	Ⅰ	Ⅰ	Ⅰ	Ⅰ	Ⅰ	Ⅰ	Ⅰ	Ⅰ	Ⅰ	ToxoDB #10

## Data Availability

Not applicable.

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
