# Peer review of "First Molecular Detection and Genotype Identification of Toxoplasma gondii in Chickens from Farmers’ Markets in Fujian Province, Southeastern China"

_pathogens, 2023, doi:10.3390/pathogens12101243_

Round 1
Reviewer 1 Report
This study evaluated the prevalence and genotypes of Toxoplasma gondii in chickens from farmers' markets in Fujian Province, southeastern China. This study is interesting. Some minor issues should be clarified
1. In general, the English language of the MS is good enough but needs improvement in grammar, sentence structure and other language issues.
For example, at line 22, “32 positive DNA” should be changed to “Thirty-two positive DNA”.
2. It is suggested that the title be changed to “First Molecular detection and genotype identification of Toxoplasma gondii in chickens from farmers' markets in Fujian Province, southeastern China”.
3. The second and subsequent appearance of the Latin name of a species should be abbreviated, such as T.gondii. For example, at lines 12 and 180, “Toxoplasma gondii” should be changed to “T.gondii.
4. For the “Introduction” section, chickens should be mentioned as sentinels reminding the contamination of T. gondii oocysts in the soil
5. For the Figure 2, L358 and Apico should be at the top of the Figure 2j and 2l.
1. In general, the English language of the MS is good enough but needs improvement in grammar, sentence structure and other language issues.
For example, at line 22, “32 positive DNA” should be changed to “Thirty-two positive DNA”.
Author Response
Responses to the comments and suggestions of reviewer 1
Point 1:In general, the English language of the MS is good enough but needs improvement in grammar, sentence structure and other language issues. For example, at lines 22, “32 positive DNA” should be changed to “Thirty-two positive DNA”.
Response 1: We have revised “32 positive DNA” to “Thirty-two positive DNA” in the text accordingly.
Point 2:It is suggested that the title be changed to “First Molecular detection and genotype identification of Toxoplasma gondii in chickens from farmers' markets in Fujian Province, southeastern China”.
Response 2: Thank you very much for your constructive suggestions. We have revised the title of this manuscript accordingly suggestions.
Point 3:The second and subsequent appearance of the Latin name of a species should be abbreviated, such as T. gondii. For example, at lines 12 and 180, “Toxoplasma gondii” should be changed to “T. gondii.
Response 3: We have revised in the text accordingly.
Point 4: For the “Introduction” section, chickens should be mentioned as sentinels reminding the contamination of T. gondii oocysts in the soil.
Response 4: Thank you very much for your favorable suggestions. We have added relevant content in the revised manuscript. (page 2, lines 53-54)
Point 5:For the Figure 2, L358 and Apico should be at the top of the Figure 2j and 2l.
Response 5: Revised accordingly.
Reviewer 2 Report
The present paper studied the incidence of Toxoplasma infection in free-range chicken from Fujian province in China. I believe that some points should be improved or clarified before the publication of this work.
Introduction:
Line 55: Please re-written
While there have been numerous studies on T. gondii genotypes in chickens [16], the prevalence and genotype of T. gondii infection specifically in chickens from Fujian Province have not been reported to date.
What is the incidence of Toxoplasma infection in the human population of Fujian? I believe this information is very important!
Discussion:
Lines 153-156
The prevalence of T. gondii infection showed a significant difference among the various tissues of the samples. One possible explanation is that the blood cells of poultries are nucleated cells, which makes them susceptible to T. gondii invasion. The parasite can enter the host's nucleated cells through the bloodstream and lymphatic system, allowing for dissemination throughout the body via the blood cells [32].
Are authors suggesting that tissue infection variation is due to parasites infecting the red blood cells? I suggest removing it from the discussion. This affirmation has no support.
Lines 170-171: When T. gondii infects chickens, the whole body's blood will undergo gas exchange through the lungs, resulting in a high detection rate of T. gondii in the lungs 171 [32].
The cited work studied the dissemination of parasites during acute infection in mice. Again, authors should revise if speculating the route of dissemination of parasites is appropriate. This has not been studied in chicken.
Besides, is this important for this work? The idea of the present work is to show the incidence of T. gondii infection in chickens, right?
Lines 170-174: Due to the dietary habits in Fujian, most of the internal organs of chickens are consumed by people, but the lungs are not within the range of people's favorite foods. Untreated lungs can be casually discarded by people, increasing the risk of transmission of T. gondii.
Is the discard of untreated lungs a risk of transmission for who or what? Humans, stray cats, pigs, chickens...?
Minor:
Line 12: change human for humans
Line 13: However, it remains unreported the prevalence
Line 16: … lungs, AND muscles)
Line 18: Out of the 577 chickens, thirty-two (5.5%) tested positive
And so on…
The article needs meticulous English revision. The text is truncated and contains several grammatical errors.
Author Response
Responses to the comments and suggestions of reviewer 2
Point 1: Question: Lines 55: Please re-written
While there have been numerous studies on T. gondii genotypes in chickens [16], the prevalence and genotype of T. gondii infection specifically in chickens from Fujian Province have not been reported to date. What is the incidence of Toxoplasma infection in the human population of Fujian? I believe this information is very important!
Response 1: Thank you very much for your constructive suggestions. We have revised and added relevant information in the text accordingly. (page 2, lines 54-57)
Point 2: Lines 153-156: The prevalence of T. gondii infection showed a significant difference among the various tissues of the samples. One possible explanation is that the blood cells of poultries are nucleated cells, which makes them susceptible to T. gondii invasion. The parasite can enter the host's nucleated cells through the bloodstream and lymphatic system, allowing for dissemination throughout the body via the blood cells [32]. Are authors suggesting that tissue infection variation is due to parasites infecting the red blood cells? I suggest removing it from the discussion. This affirmation has no support.
Response 2: Thank you very much for your constructive suggestions. We have deleted these sentences accordingly.
Point 3: Lines 170-171: When T. gondii infects chickens, the whole body's blood will undergo gas exchange through the lungs, resulting in a high detection rate of T. gondii in the lungs [32].
The cited work studied the dissemination of parasites during acute infection in mice. Again, authors should revise if speculating the route of dissemination of parasites is appropriate. This has not been studied in chicken. Besides, is this important for this work? The idea of the present work is to show the incidence of T. gondii infection in chickens, right?
Response 3: Thank you very much for your favorable suggestions. We have deleted this sentence accordingly.
Point 4: Lines 170-174: Due to the dietary habits in Fujian, most of the internal organs of chickens are consumed by people, but the lungs are not within the range of people's favorite foods. Untreated lungs can be casually discarded by people, increasing the risk of transmission of T. gondii. Is the discard of untreated lungs a risk of transmission for who or what? Humans, stray cats, pigs, chickens...?
Response 4: Thank you very much for your constructive suggestions. We have added relevant content in our revised manuscript accordingly. (page 7, lines 187-189)
Point 5: Lines 12: change human for humans
Response 5: Revised accordingly.
Point 6: Lines 13: However, it remains unreported the prevalence
Response 6: Revised accordingly.
Point 7: Lines 16: lungs, AND muscles
Response 7: Revised accordingly.
Point 8: Lines 18: Out of the 577 chickens, thirty-two (5.5%) tested positive
Response 8: Revised accordingly.
Point 9: The article needs meticulous English revision. The text is truncated and contains several grammatical errors.
Response 9: The English grammar, sentence structure, use of vocabulary, spelling and syntax of the manuscript have undergone English language editing by MDPI. The number of Language editing certificate is 70331 and has been uploaded.
Reviewer 3 Report
This is an epidemiological study on the prevalence of T. gondii infection in free range chickens bought at different markets in the Fujian province along with genotyping of the detected parasites where possible. Despite the fact that a large number of samples was tested (577) and different organs were examined, the study is not well designed as it is impossible to ascertain how the results are presented. The authors state that 32 out of 577 animals were infected but it is impossible to ascertain how they've arrived at this result: is it because one of the organs was infected or all of them? Also, it is not possible to ascertain whether multiple organs from the same animal were infected or from different animals. Because of this, the genotyping is entirely unclear and cannot be evaluated at all, as it is not clear whether the genotyping was done on one organ per animal or multiple organs of the same animal--this radically changes the interpretation of the data! The results section are a major limitation of the study, they must be organized in a different manner and presented clearly if the manuscript is to be published. Due to the unclear presentation of the results, unfortunately, the discussion is impossible to evaluate, as many of the conclusions simply do not apply at this time--they are impossible to verify based on the presented results. In addition, the introduction must be organized better in order to clearly state what the authors are trying to present in this paper--there seem to be far too many aspects to this study, so please try to focus the manuscript, otherwise, it appears rather disorganized and the scientific significance is lost. In addition, please have the manuscript reviewed for language--the language is at times very difficult to follow and simply cannot be published in this way.
The language must be revised by someone more proficient in the English language. Parts of the manuscript are difficult to follow.
Author Response
Responses to the comments and suggestions of reviewer 3
Point 1: Question:
This is an epidemiological study on the prevalence of T. gondii infection in free range chickens bought at different markets in the Fujian province along with genotyping of the detected parasites where possible. Despite the fact that a large number of samples was tested (577) and different organs were examined, the study is not well designed as it is impossible to ascertain how the results are presented. The authors state that 32 out of 577 animals were infected but it is impossible to ascertain how they've arrived at this result: is it because one of the organs was infected or all of them? Also, it is not possible to ascertain whether multiple organs from the same animal were infected or from different animals. Because of this, the genotyping is entirely unclear and cannot be evaluated at all, as it is not clear whether the genotyping was done on one organ per animal or multiple organs of the same animal--this radically changes the interpretation of the data! The results section are a major limitation of the study, they must be organized in a different manner and presented clearly if the manuscript is to be published. Due to the unclear presentation of the results, unfortunately, the discussion is impossible to evaluate, as many of the conclusions simply do not apply at this time--they are impossible to verify based on the presented results. In addition, the introduction must be organized better in order to clearly state what the authors are trying to present in this paper--there seem to be far too many aspects to this study, so please try to focus the manuscript, otherwise, it appears rather disorganized and the scientific significance is lost. In addition, please have the manuscript reviewed for language--the language is at times very difficult to follow and simply cannot be published in this way.
Response 1: Thank you very much for your constructive suggestions. We have been modified the result section and replaced the data of Table 1, 2 and 3 in the revised manuscript accordingly.
- In the present study, according to the following formula to ascertain infection rate of T. gondii: Prevalence of infected chickens (%) = 100 × number of PCR-positive chickens regarding at least one tissue/total number of tested chickens. (page 3, lines 102-104)
- We have performed statistical analyses of multiple tissues or organs from different animals according to different regions. The PCR results of 128 tissue and organ samples of the thirty-two T. gondii-positive chickens from different regions showed the presence of multiple tissues or organs infected in the same animal. (page 3, lines 125-133; page 4, Table 2)
- We focused on genotyped multiple tissues or organs from different T. gondii-positive chickens. The results found that lung samples of five chickens from Quanzhou (QZC68), Zhangzhou (ZZC18) and Sanming (SMC20, SMC22 and SMC30), one heart sample of chicken from Zhangzhou (ZZC19) and one liver sample of chicken from Sanming (SMC31) were successfully amplified at 10 genetic markers, respectively. The genotypes were all identified as Type I (ToxoDB #10) upon comparison with the T. gondii database. (page 4, lines 136-142; page 6, Table 3).
- The English grammar, sentence structure, use of vocabulary, spelling and syntax of the manuscript have undergone English language editing by MDPI. The number of Language editing certificate is 70331 and has been uploaded.
Round 2
Reviewer 2 Report
The authors addressed all raised concerns.
Author Response
Thank you for your diligent review of our manuscript, and we appreciate your approval.
Reviewer 3 Report
This is a revised version of a manuscript regarding the prevalence, tissue distribution and genotypes of T. gondii in free range chickens in China. The experimental design is sound, yet the data are not clearly presented again and the discussion needs to be improved. The presentation of the data needs to reflect the overall aim of the manuscript but here it is not clear whether the authors are particularly interested in epidemiology (prevalence) or tissue distribution and/or genotypes, as the discussion covers all and is not particularly clear in regard to the significance of each result. Moreover, the authors need to take into account inherent limitations of the methodology used here--the tissue isolation kit they have used is only capable of processing up to 200mg of tissue, yet they've not described how they've actually collected the tissue for extraction and/or whether entire organs were actually extracted. This will have major implications of the final tissue distribution results. In addition, the genotyping data has to be discussed in much more detail than provided here, as it is quite remarkable due to the methodology used and the results obtained. It is a bit of a concern that the authors have not emphasized that at all in the discussion.
Line by line:
L83: There needs to be some information regarding the extraction itself: how much tissue did you use and how was the selection made. Please also state if you used more than one piece of tissue per each organ. This will explain your tissue distribution results and please understand that detection in muscle tissue using the method you have used is generally non-informative. There are other methods (magnetic capture PCR) which have been developed to overcome the issue of large tissues when it comes to detection of T. gondii.
L102: Please check the syntax of this sentence.
L134: Table 2. The percentages here are misleading because you have very different numbers of starting samples for each region and also different numbers of positive samples. Please consider removing the percentages as you have few animals which were positive in particular regions and maybe even removing the regions entirely, the regional prevalence is already shown in the previous result. This table is not particularly clear in terms of the significance of results. As mentioned, tissue distribution results are expected to be affected by the inherent limits of the methodology.
L142: You do not need the database to identify the genotype if you have reference standards (which you do).
L150: Table 3. These are not isolates. These are strains you've detected by PCR. Please understand the tremendous difference here. And this is something that needs to be mentioned in the discussion section as well.
L154: While it is true that chronic toxoplasmosis doesn't usually have clinical manifestations in chickens, you have no information and/or data of any sort to know whether the birds you've tested were in fact chronically infected. Please understand that this is a misleading sentence here, although it is correct. Consider deleting and/or giving the paragraph a different context.
L161-163: What is the significance of this sentence?
L172-173: What is the significance of this sentence?
L174: Improper terminology, please use DNA.
L174-184: Please note: Most of the references describe experimental infections of chickens. You're working with natural infection. Please make a note of the difference--tissue distribution will very much depend on the time post infection (which you do not know in this case) and whether T. gondii was in the tachyzoite, tissue cyst or oocyst form at the time of infection (which you again do not know in this case). So distribution needs to be given special attention in the discussion.
L187: Tissue cysts do not contaminate the environment, especially not water! Please revise the sentence, it is incorrect.
L191: As far as I am aware, there is currently no mandate or even specific regulation for T. gondii inspection in meat. Please be more specific what you mean here.
L202: Please note that it is well known that type I (ToxoDB#10) can be transmitted across different species, your data however does not show that at all, so please be very specific what it is that you would like to state here.
L203-204: Stating that type I is common in China and claiming that there may be a common source of infection is contradictory.
L205-210: This is a conjecture and should be removed. Please stick to the significance of your data and not a hypothetical situation.
L195-212: Please have another look at the arguments you are trying to make in this paragraph. It is not a very well formulated paragraph at all, it is contradictory and generally unclear. There is very little actual information discussed which pertains to your own data. Besides, finding such an overwhelming dominance of type I (21%) needs to be discussed in light of other data from China, as this is a very high frequency of type I, not just for China but in fact for most other places where the population structure of T. gondii has been investigated. Finally, please understand that genotyping from isolates and direct genotyping (which is what you have done) is tremendously different technically. Your data in fact suggest that the abundance of T. gondii DNA in the tissues was very high to yield a full PCR-RFLP genotype for 7 samples--that is pretty remarkable.
Please check individual sentences once again, there are some minor mistakes to correct. Please pay special attention to the syntax.
Author Response
Response to Reviewer 3 Comments
Point 1: This is a revised version of a manuscript regarding the prevalence, tissue distribution and genotypes of T. gondii in free range chickens in China. The experimental design is sound, yet the data are not clearly presented again and the discussion needs to be improved. The presentation of the data needs to reflect the overall aim of the manuscript but here it is not clear whether the authors are particularly interested in epidemiology (prevalence) or tissue distribution and/or genotypes, as the discussion covers all and is not particularly clear in regard to the significance of each result. Moreover, the authors need to take into account inherent limitations of the methodology used here--the tissue isolation kit they have used is only capable of processing up to 200mg of tissue, yet they've not described how they've actually collected the tissue for extraction and/or whether entire organs were actually extracted. This will have major implications of the final tissue distribution results. In addition, the genotyping data has to be discussed in much more detail than provided here, as it is quite remarkable due to the methodology used and the results obtained. It is a bit of a concern that the authors have not emphasized that at all in the discussion.
Response 1: Thank you very much for your constructive suggestions. We have modified and revised the manuscript accordingly.
Point 2: L83: There needs to be some information regarding the extraction itself: how much tissue did you use and how was the selection made. Please also state if you used more than one piece of tissue per each organ. This will explain your tissue distribution results and please understand that detection in muscle tissue using the method you have used is generally non-informative. There are other methods (magnetic capture PCR) which have been developed to overcome the issue of large tissues when it comes to detection of T. gondii.
Response 2: Thank you very much for your constructive suggestions. I made a mistake to descripted the name of DNA extracted kit. In the present study, actually, the total DNA of tissue per each organ were obtained using the commercial E.Z.N.A® Tissue DNA kit (Omega Biotek Inc., Norcross, GA, USA). We have revised in the text. (page 3, lines 88-91)
Point 3: L102: Please check the syntax of this sentence.
Response3: Revised accordingly.
Point 4: L134: Table 2. The percentages here are misleading because you have very different numbers of starting samples for each region and also different numbers of positive samples. Please consider removing the percentages as you have few animals which were positive in particular regions and maybe even removing the regions entirely, the regional prevalence is already shown in the previous result. This table is not particularly clear in terms of the significance of results. As mentioned, tissue distribution results are expected to be affected by the inherent limits of the methodology.
Response 4: Thank you very much for your constructive suggestions. We have revised the data in Table 2 accordingly. The main purpose of this table is to highlight T. gondii infection rates in the different tissue and organ of T. gondii-positive chickens in each region.
Point 5: L142: You do not need the database to identify the genotype if you have reference standards (which you do).
Response 5: Revised accordingly.
Point 6: L150: Table 3. These are not isolates. These are strains you've detected by PCR. Please understand the tremendous difference here. And this is something that needs to be mentioned in the discussion section as well.
Response 6: Thank you very much for your valuable suggestions. We have revised Table 3 accordingly.
Point 7: L154: While it is true that chronic toxoplasmosis doesn't usually have clinical manifestations in chickens, you have no information and/or data of any sort to know whether the birds you've tested were in fact chronically infected. Please understand that this is a misleading sentence here, although it is correct. Consider deleting and/or giving the paragraph a different context.
Response 7: Thank you very much for your constructive suggestions. We have deleted these sentences accordingly.
Point 8: L161-163: What is the significance of this sentence?
Response 8: We have deleted this sentence accordingly.
Point 9: L172-173: What is the significance of this sentence?
Response 9: We have deleted this sentence accordingly.
Point 10: L174: Improper terminology, please use DNA.
Response 10: Revised accordingly.
Point 11: L174-184: Please note: Most of the references describe experimental infections of chickens. You're working with natural infection. Please make a note of the difference--tissue distribution will very much depend on the time post infection (which you do not know in this case) and whether T. gondii was in the tachyzoite, tissue cyst or oocyst form at the time of infection (which you again do not know in this case). So distribution needs to be given special attention in the discussion.
Response 11: Thank you very much for your constructive suggestions. We have added the related information in our revised manuscript accordingly.
Point 12: L187: Tissue cysts do not contaminate the environment, especially not water! Please revise the sentence, it is incorrect.
Response 12: We have revised this sentence accordingly.
Point 13: L191: As far as I am aware, there is currently no mandate or even specific regulation for T. gondii inspection in meat. Please be more specific what you mean here.
Response 13: Thank you very much for your valuable suggestions. We have revised this sentence in the manuscript accordingly.
Point 14: L202: Please note that it is well known that type I (ToxoDB#10) can be transmitted across different species, your data however does not show that at all, so please be very specific what it is that you would like to state here.
Response 14: Thank you very much for your constructive suggestions. We have added and cited many articles in relation to T. gondii genotype identification from different host to illustrate that type I (ToxoDB#10) can spread between different species.
Point 15: L203-204: Stating that type I is common in China and claiming that there may be a common source of infection is contradictory.
Response 15: We have deleted this sentence in our revised manuscript.
Point 16: L205-210: This is a conjecture and should be removed. Please stick to the significance of your data and not a hypothetical situation.
Response 16: We have deleted these sentences accordingly.
Point 17: L195-212: Please have another look at the arguments you are trying to make in this paragraph. It is not a very well formulated paragraph at all, it is contradictory and generally unclear. There is very little actual information discussed which pertains to your own data. Besides, finding such an overwhelming dominance of type I (21%) needs to be discussed in light of other data from China, as this is a very high frequency of type I, not just for China but in fact for most other places where the population structure of T. gondii has been investigated. Finally, please understand that genotyping from isolates and direct genotyping (which is what you have done) is tremendously different technically. Your data in fact suggest that the abundance of T. gondii DNA in the tissues was very high to yield a full PCR-RFLP genotype for 7 samples--that is pretty remarkable.
Response 17: Thank you very much for your constructive suggestions. We have revised and discussed the related content accordingly. (page 9, lines 202-209)
Round 3
Reviewer 3 Report
Thank you for making the necessary revisions. Please now revise the abstract to correspond to the revisions done in the rest of the manuscript. In particular, L22-23, please revise to indicate that this represents 7 chickens and also please do mention all of your results in the discussion, as the significant differences found between the different regions you've sampled--this is not considered at all, you simply mention the difference with regard to other provinces.
Language is much improved. Minor changes are to be made throughout the paper.
Author Response
Response to Reviewer 3 Comments
Point 1: Thank you for making the necessary revisions. Please now revise the abstract to correspond to the revisions done in the rest of the manuscript. In particular, L22-23, please revise to indicate that this represents 7 chickens and also please do mention all of your results in the discussion, as the significant differences found between the different regions you've sampled--this is not considered at all, you simply mention the difference with regard to other provinces
Response 1: Thank you very much for your constructive suggestions. We have revised and added the related content accordingly.